# Enantioselectivity of Pentedrone and Methylone on Metabolic Profiling in 2D and 3D Human Hepatocyte-like Cells

**DOI:** 10.3390/ph15030368

**Published:** 2022-03-17

**Authors:** Bárbara Silva, Joana Saraiva Rodrigues, Ana Sofia Almeida, Ana Rita Lima, Carla Fernandes, Paula Guedes de Pinho, Joana Paiva Miranda, Fernando Remião

**Affiliations:** 1Associate Laboratory i4HB—Institute for Health and Bioeconomy, Faculdade de Farmácia, Universidade do Porto, Rua Jorge Viterbo Ferreira, 228, 4050-313 Porto, Portugal; ritacmlima@hotmail.com (A.R.L.); pguedes@ff.up.pt (P.G.d.P.); 2UCIBIO-REQUIMTE, Laboratório de Toxicologia, Departamento de Ciências Biológicas, Faculdade de Farmácia, Universidade do Porto, Rua Jorge Viterbo Ferreira, 228, 4050-313 Porto, Portugal; 3Laboratório de Química Orgânica e Farmacêutica, Departamento de Ciências Químicas, Faculdade de Farmácia, Universidade do Porto, Rua de Jorge Viterbo Ferreira, 228, 4050-313 Porto, Portugal; up201605043@edu.ff.up.pt (A.S.A.); cfernandes@ff.up.pt (C.F.); 4Research Institute for Medicines (imed), Faculdade de Farmácia, Universidade de Lisboa, Avenida Professor Gama Pinto, 1649-003 Lisboa, Portugal; joana.s.rodrigues@campus.ul.pt (J.S.R.); jmiranda@ff.ulisboa.pt (J.P.M.); 5Centro Interdisciplinar de Investigação Marinha e Ambiental (CIIMAR), Universidade do Porto, Avenida General Norton de Matos, 4450-208 Matosinhos, Portugal

**Keywords:** cathinones, chirality, enantiomers, liver, metabolism, 3D-HLCs

## Abstract

Pentedrone and methylone can express stereoselectivity in toxicokinetic and toxicodynamic processes. Similarly, their chiral discrimination in metabolism, which was not yet evaluated, can result in different metabolic profiles and subsequent hepatotoxic effects. Therefore, the aim of this work was to assess, for the first time, both the hepatic cytotoxic and metabolic profile of pentedrone and methylone enantiomers using physiologically relevant in vitro models. The hepatotoxicity of these compounds was observed in a concentration-dependent manner in human stem-cell-derived hepatocyte-like cells (HLCs) cultured under 3D (3D-HLCs) and 2D (2D-HLCs) conditions. Enantioselectivity, on the other hand, was only shown for pentedrone (1 mM) in 3D-HLCs, being *R*-(−)-pentedrone the most cytotoxic. Furthermore, the metabolic profile was initially evaluated in human liver microsomes (HLM) and further demonstrated in 3D-HLCs and 2D-HLCs applying a gas chromatography coupled to a mass spectrometer (GC–MS) technique. Methylone and pentedrone showed distinct and preferential metabolic routes for their enantiomers, resulting in the production of differentiated metabolites; *R*-(+)-methylone and *R*-(−)-pentedrone are the most metabolized enantiomers. In conclusion, the results demonstrated enantioselectivity for pentedrone and methylone in the metabolic processes, with enantioselectivity in cytotoxicity for pentedrone.

## 1. Introduction

During the last decade, the consumption of synthetic cathinones for recreation experiences has gained popularity although associated with several toxic effects, which constitutes a public health concern [1]. Among the most consumed cathinone derivatives in the USA and Europe are pentedrone and methylone [2,3]. All cathinone derivatives contain a stereogenic center—as chiral molecules, their toxicokinetic and toxicodynamic characteristics can be different for each enantiomer [4]. The toxicological effects of a drug depend mostly on its interaction with biomacromolecules, such as proteins, metabolic enzymes, and cellular receptors. As many of these biological molecules in human organisms comprise intrinsic chirality [5], the enantiomers of a chiral drug may display differences in their toxicokinetic properties and/or toxicological activities as a result of chiral discrimination during these processes [6]. In fact, our group already demonstrated the enantioselectivity of pentedrone and methylone in some toxicokinetic processes such as the intestinal absorption, using Caco-2 cell monolayer, with *R*-(−)-pentedrone and *S*-(−)-methylone the most permeable compounds [7]. Similarly, differentiated cytotoxic profiles were demonstrated in a neuronal in vitro model (SH-SY5Y cells), where the *S*-(+)-pentedrone and *R*-(+)-methylone showed the most cytotoxic effects [8].

Drug metabolism studies, on the other hand, provide crucial information to facilitate the understanding of the pharmacological and toxicological action of drugs. The stereoselectivity in metabolism often results in the biotransformation of enantiomers at different rates and/or different routes, being considered the primary factor responsible for the differences observed in enantioselective drug disposition [6]. The main route of administration for these compounds is ingestion, making the liver a target organ of their toxicity. Indeed, clinical data have established liver damage as an important problem of synthetic cathinone abuse [1,9]. Indeed, in vitro studies, using primary rat hepatocytes (PRH), showed cytotoxic effects of racemic methylone and pentedrone in a concentration-dependent manner, with an EC_50_ of 1.6 and 0.7 mM, respectively [10]. The same study also demonstrated a common mechanism for pentedrone and methylone hepatic damage involving oxidative stress (ROS/RNS increase and GSH decrease) and mitochondrial dysfunction but failed to fully clarify the role of the cathinones’ metabolism in the observed hepatotoxicity. The metabolism of methylone is well described, using several in vitro and in vivo models as represented in Figure 1. Methylone can undergo *N*-demethylation, *O*-demethylation, *N*-oxidation, and reduction of *β*-keto moiety. The dihydroxy metabolites may also undergo *O*-methylation by catechol-*O*-methyltransferase (COMT) [11,12,13,14]. Studies in human liver microsomes (HLM) proved that methylone is initially metabolized by CYP2D6, with minor contributions of CYP1A2, CYP2B6, and CYP2C19 [15]. Regarding pentedrone, Uralets et al. [2] found, in human urine of patients exposed to pentedrone, metabolites resulting from *N*-demethylation and *β*-ketone reduction followed by *N*-demethylation (Figure 2). Nonetheless, the role of pentedrone and methylone enantioselectivity on toxicity and metabolism in the liver has not yet been studied. Thus, the metabolic profile of each enantiomer should be evaluated, namely using human physiologically relevant (advanced) in vitro hepatic models.

The development of advanced human in vitro hepatic models for the prediction of hepatotoxicity remains a challenge in the medicinal chemistry and pharmaceutical industry. Primary human hepatocytes (PHH) are the gold standard for hepatotoxicity and drug metabolism screening. However, their limited availability and progressive loss of functionality in culture [17], grants stem-cell-derived hepatic human models a great potential to overcome the limitations of the current in vitro models, in terms of function, stability, and competence [18,19,20]. Traditionally, such cultures are carried out under adherent conditions wherein the cells are attached to a glass or plastic dish, designated as 2D culture models. The advantages of 2D cultures are several and are mostly associated with simple and low-cost maintenance of the cell culture and with the performance of functional tests [17,18,19,20]. However, 2D models also have numerous disadvantages. Adherent cell cultures do not mimic the natural structures of tissues, namely cell–cell and cell–extracellular matrix (ECM) interactions responsible for cell differentiation, proliferation, vitality, expression of genes and proteins, responsiveness to stimuli, drug metabolism, and other cellular functions. Therefore, under such conditions cell morphology and phenotype are altered, affecting their function, structural organization, secretion, and cell signaling [17,18,19,20]. Owing to the many disadvantages of 2D systems, major efforts have been made to provide more physiologically relevant models for nonclinical hepatotoxicity assessment. Three-dimensional models are more complex culture systems where multicellular microphysiological devices are explored within a vision to replicate the characteristics and response of human tissues in vivo. Three-dimensional culture systems enable cell polarization, the formation of nutrient and oxygen gradients, and higher cell–cell and cell–ECM contacts, which are key conditions for the maintenance of hepatic-specific functions [18,19,20]. We previously developed a hepatic differentiation protocol for obtaining hepatocyte-like cells (HLCs) from human mesenchymal stem cells (hnMSCs) [21,22,23]. The hnMSCs are a promising stem-cell type, regarding accessibility, expandability, ethical, and safety concerns, when compared to human-induced pluripotent stem cells (hiPSCs) or human embryonic stem cells (hESCs). The hepatic phenotype obtained was confirmed through an unbiased whole genome analysis, being placed between PHH and HepG2, and distant from hnMSCs, with stable and inducible biotransformation activity [24]. Importantly, when adapted to 3D culture conditions, HLCs phenotype and functionality was improved, being able to mimic the biotransformation profiles of drugs [19,25]. Therefore, 3D-cultured HLCs may represent an important tool to build more reliable human in vitro models for the assessment of potential toxic effects. 

Thus, the aim of this work was to evaluate, for the first time, the hepatic cytotoxic and metabolic profiles of pentedrone and methylone enantiomers. After the enantiomers’ purification by liquid chromatography (LC) using a Chiralpak AS^®^ column [26], metabolic studies were performed using HLM, which enabled metabolite identification and their relative quantification by gas chromatography coupled to a mass spectrometer (GC–MS). Finally, the enantiomers’ cytotoxic and metabolic profiles were evaluated using HLCs cultured under 3D (3D-HLCs) and 2D (2D-HLCs) in vitro conditions that further allowed the confirmation of metabolite formation and underlying enantioselectivity.

## 2. Results

### 2.1. Racemic Pentedrone and Methylone Compounds Are Cytotoxic to HLCs Cells in a Concentration-Dependent Manner

Herein, we evaluated the cytotoxic profile of methylone and pentedrone using 3D- and 2D-HLCs cultures through cell viability assay and cell morphology (Figure 3 and Figure 4). 

According to the results obtained, a concentration-dependent hepatotoxic effect was observed in all culture systems for both drugs after 24 h of exposure to pentedrone and methylone racemates (0–20.0 mM) (Figure 3). However, when compared to control cells, pentedrone and methylone exhibited significant cytotoxicity for all concentrations tested in 2D-HLCs; whereas, in 3D-HLCs, cytotoxicity was only observed for concentrations higher than 1.0 and 3.0 mM for pentedrone and methylone, respectively (Figure 3). Indeed, the EC_50_ values estimated for pentedrone were 2.8 mM (2D) and 3.6 mM (3D) and for methylone were 4.8 mM (2D) and 5.2 mM (3D) (Figure 3), revealing higher EC_50_ values in 3D cultures than in 2D cultures. Finally, in this 2D/3D comparative study, the phenotypic morphological changes were also monitored at the end of the assay, as shown in Figure 4. In accordance with the previous results, cytotoxicity through cell detachment was observed in 2D-HLCs with lower concentrations, while spheroids, 3D-HLCs, started to lose cells in the outer borders, developing an irregular shape, at higher concentrations (Figure 4).

### 2.2. Enantioselectivity Is Observed in Pentedrone and Methylone Cytotoxicity

Cytotoxic enantioselectivity studies induced by these drugs were performed using the lower observed cytotoxic concentrations for 3D-HLCs, as calculated through the analysis of the racemate concentration–response curves, 0.5 and 1 mM. As expected, for the concentrations tested, higher cytotoxicity levels were observed in 2D-HLCs. Interestingly, enantioselectivity was only observed for pentedrone enantiomers at 1 mM in 3D cultures, with *R*-(−)-pentedrone being the most cytotoxic enantiomer (Figure 5).

### 2.3. Pentedrone and Methylone Show Enantioselective Metabolic Profiles

#### 2.3.1. Identification of Pentedrone and Methylone Metabolites

Given their chiral nature, studying the metabolic profile of cathinone’s enantiomers is crucial to understand their toxicity profile. Herein, in a first step, the metabolism assays were performed with HLMs, which enabled the development of a GC–MS methodology using specific *m*/*z* (SIM mode), for the identification of pentedrone and methylone phase I metabolites. It further permitted a first insight into possible differences between the enantiomers metabolism prior to the metabolism studies with the HLCs in vitro models. The GC–MS analysis of the HLMs metabolic assay samples enabled the identification of four metabolites of methylone (*N*-hydroxymethylone, DHMC, dihydromethylone, and nor-methylone) and two metabolites of pentedrone (dihydropentedrone and nor-pentedrone). It also enabled the establishment of the retention times (RT) for *N*-hydroxymethylone, DHMC, dihydromethylone, and nor-methylone: 8.45, 8.95, 9.50, and 9.86 min, respectively. The RT for dihydropentedrone and nor-pentedrone were 7.14 and 7.76, respectively. The IS (methcathinone) retention time was 6.46 min. The GC conditions for the determination of *N*-hydroxymethylone, DHMC, dihydromethylone, and nor-methylone resulted in well-resolved peaks eluting in less than 11 min (Figure 6). Regarding pentedrone metabolites (nor-pentedrone and dihydropentedrone), their determination was possible in 9 min (Figure 6). 

The identification of each metabolite peak was assessed analyzing the chromatograms acquired by SIM mode and comparing the abundance of the common *m*/*z* of methylone with *m*/*z* of methylone metabolites and between the common *m*/*z* of pentedrone and *m*/*z* of pentedrone metabolites. For example, *m*/*z* 121, 149, 170, 198, and 319 were defined for *N*-hydroxymethylone. This metabolite have *m*/*z* 121, 149, and 170 in common with the parent compound (methylone). Therefore, these three ions were used to make the comparison of the spectrum (Figure 7) for each chromatographic peak. This procedure was carried out for all the metabolites (data not shown). Moreover, the samples of microsomes T = 0 min and T = 60 min were analyzed whenever needed to clarify the correct peak, to ensure that the metabolite did not exist at time 0.

To validate this identification hypothesis (performed by the mass spectra comparison), the dihydromethylone and dihydropentedrone metabolites were synthetized (using sodium borohydride strategy; 64–96% yields) [27] by keto group reduction of methylone and pentedrone, respectively, to the corresponding amino alcohols. The synthetic strategy using sodium borohydride [27] offered good yields (64–96%). The reduced compounds were then analyzed by GC–MS after TFAA derivatization, and the fragmentation compared with the theoretical one. Figure 8 shows that the RT and mass spectrum of the standard matches the predicted RT for the metabolite dihydropentedrone.

Concerning the metabolic profiles, different metabolite formation rates were detected for methylone (Figure 9A and Appendix A) and pentedrone (Figure 9B and Appendix A) enantiomers. Importantly, these results enabled setting up a methodology for the identification of methylone/pentedrone metabolites, while indicating pentedrone and methylone enantioselectivity for the phase I metabolic profile. As such, the same methodology was then applied to evaluate pentedrone and methylone enantioselectivity in phase I metabolism in 2D and 3D-HLCs. 

#### 2.3.2. Enantioselectivity Metabolism Profiling Studies in 2D- and 3D-HLCs

In both 2D and 3D-HLC culture systems, preferential metabolism routes for each enantiomer were observed, resulting in differentiated profiles in metabolite formation (Figure 10). Nevertheless, regarding methylone, *N*-Hydroxymethylone was the metabolite that showed the highest formation both in 2D-HLCs and 3D-HLCs, being more pronounced in the 3D model. On the other hand, regarding pentedrone, the metabolite that showed a more pronounced formation in both systems was dihydropentedrone, being also more accentuated in the 3D model.

Moreover, concerning the methylone metabolism profile, in 2D-HLCs, *S*-(−)-methylone was preferentially metabolized via *N*-demethylation, resulting in a high production of nor-methylone in relation to *R*-(+)-methylone, whereas *R*-(+)-methylone appeared to be preferentially metabolized by *O*-demethylation and *N*-hydroxylation, resulting in a more pronounced production of DHMC and *N*-hydroxymethylone in cells exposed to *R*-(+)-methylone compared to cells exposed to *S*-(−)-methylone. Regarding the *β*-keto reduction, no enantioselectivity was observed (Figure 10A). In 3D-HLCs, *R*-(+)-Methylone was preferentially metabolized by all metabolic routes, except for *O*-demethylation (Figure 10B). Cells exposed to *R*-(+)-methylone had a more pronounced production of nor-methylone (*N*-demethylation), dihydromethylone (*β*-keto reduction), and *N*-hydroxymethylone (*N*-hydroxylation) compared to cells exposed to *S*-(−)-methylone. No enantioselectivity was observed regarding the *O*-demethylation path. Regarding the pentedrone metabolic profile, in 2D-HLCs, *R*-(−)-pentedrone was preferentially metabolized by all metabolic routes in relation to *S*-(+)-pentedrone (Figure 10C), with a more pronounced production of nor-pentedrone (*N*-demethylation) and dihydropentedrone (*β*-keto reduction). Regarding the extracted samples from 3D-HLCs (Figure 10D), *R*-(−)-pentedrone was preferentially metabolized by *β*-keto reduction, resulting in a more pronounced dihydropentedrone formation compared to *S*-(+)-pentedrone. Enantioselectivity was not observed regarding the *N*-demethylation.

## 3. Discussion

Despite their importance, enantioselectivity studies still constitute a significant challenge for the research community, owing to the difficulties in enantiomer isolation and to the lack of data in relevant in vitro models. Liver’s susceptibility to the toxicity of synthetic cathinones has already been proved by clinical data [1,9] and in vitro studies for different cathinone derivatives, namely for pentedrone, methylone, 4-MEC [10], 3,4-methylenedioxypyrovalerone (MDPV) [10,28], butylone, buphedrone, and 3,4-DMMC [29] in PRH. However, the enantioselectivity of these compounds has not yet been studied in the liver. Thus, the present work aimed to unravel the enantioselectivity metabolic and cytotoxic profile of pentedrone and methylone by resorting to two in vitro hepatic models, HLCs cultured in 2D and 3D. Herein, it is shown that the estimated EC_50_ values for both pentedrone and methylone are distinct in both models, being higher in 3D cultures than in 2D cultures. Most importantly, the enantioselectivity metabolic profile of pentedrone and methylone in 2D- and 3D-HLCs is demonstrated for the first time. Finally, it is further shown that enantioselective cytotoxicity was only observed in the 3D model, where the *R*-(−)-pentedrone, which is preferentially metabolized over the *S* enantiomer, is also the most toxic enantiomer in this cellular model.

The cytotoxicity studies of pentedrone and methylone revealed higher EC_50_ values in 3D-HLCs than in 2D-HLCs, which seem to indicate that 3D-HLCs would be able to detoxify better, probably by a higher competence in phase II reactions. Indeed, previous studies demonstrated that 3D-HLCs showed higher biocompetence in producing phase II nevirapine metabolites and upregulating phase II enzymes [25]. The EC_50_ in the mM range was also assessed for these cathinones in other hepatic models [10] and for other cathinones [29]. Noteworthy, for the same drugs, a higher cytotoxicity was observed in a neuronal model (SH-SY5Y) with an EC_50_ of 3 mM for both cathinones, whereas no cytotoxicity was observed in an intestinal model (Caco-2) [7,8]. These findings show a higher cytotoxicity of pentedrone and methylone to neuronal cells, which was also ascribed to other synthetic cathinones [30,31]. Besides the micromolar levels found in blood samples from ‘bath salts’-related intoxications [32], the liver is exposed to much higher drug concentrations than those found in the blood. In fact, it was reported that pentedrone and methylone reach liver-to-blood ratios up to 11 and 3, respectively [33,34,35]. The enantioselectivity studies on the cytotoxicity induced by these drugs were performed using 0.5 and 1 mM. Regarding their cytotoxicity and according to the results, enantioselectivity was observed only for pentedrone at 1 mM in 3D-HLCs, with *R*-(−)-pentedrone being the most cytotoxic enantiomer. Other studies performed by our group have previously demonstrated that both pentedrone and methylone present enantioselectivity in their cytotoxicity and the underlying mechanisms in SH-SY5Y cells [8]. Additionally, stereoselective permeability was observed for pentedrone and methylone in Caco-2 cells [7].

The enantioselective metabolism of chiral drugs catalyzed by phase I and II enzymes is believed to be responsible for the main stereoselectivity observed in pharmacokinetics and drug disposition [36]. The large availability of HLM, their easy use, and the fact that they are one of the best-characterized in vitro systems for drug biotransformation research contribute to the popularity of this in vitro model [37]. HLM incubations have been quite successfully used for the generation of metabolites for synthetic cathinones [15,38,39,40]. Thus, the identification and enantioselective formation of methylone and pentedrone metabolites were initially studied in ultra-pool HLM by incubating individually each enantiomer. This model enabled setting up a newly developed methodology (GC–MS based) for the identification of the phase I metabolite profile of the pentedrone and methylone enantiomers and suggested a different metabolism rate for each enantiomer. Specific metabolites of methylone and pentedrone already identified in previous studies [2,15] were selected to postulate their mass fragmentation according to their chemical structures (please see Section 4.6 in Materials and Methods description) to search only chromatographic peaks with specific ions (*m*/*z*) of these compounds and use these *m*/*z* values in SIM mode. The use of SIM mode allowed the identification of the metabolite’s peaks and allowed a better integration, which is especially important for small peaks.

As mentioned above, HLCs represent a hepatic in vitro model closer to the physiological system in functionality, stability, and competence [18,20,24]. Therefore, both 2D and 3D-HLCs were used in this work for interpreting the enantioselectivity of these compounds in metabolism after the identification of the metabolites formed in HLMs. Reaching a compromise between low toxic concentration (below the EC_30_) and a concentration that presents enantioselectivity in its action, the concentration of 1 mM was chosen for the metabolic studies. All the metabolites were determined in both HLCs models. Importantly, methylone and pentedrone showed preferential metabolic routes for their enantiomers, resulting in a differentiated metabolite production.

In the 2D model, *S*-(−)-methylone was preferentially metabolized via *N*-demethylation, whereas *R*-(+)-methylone appeared to be preferentially metabolized by *O*-demethylation and *N*-hydroxylation. Regarding 3D models, methylone showed preferential metabolism routes for their enantiomers. *R*-(+)-methylone was preferentially metabolized by all metabolic paths, except for *O*-demethylation. The *N*-hydroxylation was the preferential pathway for methylone in both models. In both models, this metabolite was preferentially produced by *R*-(+)-methylone, although its formation was more evident in the 3D model (approximately the double, as shown in Figure 10).

The *R*-(−)-pentedrone was preferentially metabolized by all metabolic routes in both models. The *β*-reduction was the preferential pathway for pentedrone in both models, with the same preference for the *R*-(−)-enantiomer. Once again, the metabolite formation was more evident in the 3D model (approximately the double, as shown in Figure 10). Furthermore, the *β*-keto reduction of pentedrone seems to produce metabolites more toxic than the parent drug. Indeed, *R*-(−)-pentedrone is preferentially metabolized by this pathway and also showed to be the most toxic enantiomer.

The metabolic profile obtained using the two models lead to different results, indicating a different metabolic profile between HLCs cultured in monolayer and spheroids, the 3D model apparently being more suitable for metabolic assays as it demonstrated better competence in detoxifying the studied cathinones.

## 4. Materials and Methods

### 4.1. Reagents and Samples

Methylone and pentedrone were kindly received from Judiciary Police and were characterized by mass spectrometry according to a method previously developed [41]. Methcathinone [42] was used as an internal standard. The enantiomers of pentedrone and methylone were isolated by LC on a Chiralpak AS^®^ (250 × 10 mm i.d., 5 μm particle size) column and fully characterized by electronic circular dichroism (ECD), polarimetry, and GC–MS, as previously described [26].

Acetonitrile, ethyl acetate, methanol, and trifluoracetic anhydride (TFAA) were purchased from Sigma-Aldrich (St. Louis, MO, USA). Purified water was obtained from a Millipore Synergy UV water purification system (Millipore A/S, Copenhagen, Denmark). All culture media and supplements, solvents (all of analytical grade) and other chemicals were acquired from Sigma-Aldrich (Madrid, Spain) unless specified. Ultra-pool human liver microsomes 150 and NADPH regeneration system solution A and B were purchased from Corning (Corning, New York, NY, USA). Solution A contained 26 mM NADP^+^, 66 mM glucose-6-phosphate, and 66 mM MgCl_2_ in H_2_O. Solution B consisted of 40 U/mL glucose-6-phosphate dehydrogenase in 5 mM sodium citrate.

#### Multi-Milligram Resolution of Pentedrone and Methylone Enantiomers

The enantiomers of pentedrone and methylone were isolated by LC on a Chiralpak AS^®^ (250 × 10 mm i.d., 5 μm particle size) column, as previously described [26]. Briefly, multiple injections (500 μL loop) using Hex:2-PrOH as mobile phase were performed. Analyses were performed at 25 °C, in isocratic mode under UV detection (254 nm). The detection parameters were adjusted to a peak start and end threshold of 200 mAU for pentedrone enantiomers and 100 mAU for methylone. The fractions of each enantiomer obtained were precipitated with HCl on diethyl ether (2 M). Working solutions for each enantiomer of pentedrone and methylone were prepared at a concentration of 0.1 mg/mL and injected, in triplicate, on the analytical Chiralpak AS-H^®^ (15 cm × 4.6 mm i.d., 5 μm particle size) column under optimized chromatographic conditions, as previously described [28], to measure their enantiomeric purity (e.r.). The enantiomeric ratio (ER) was determined by the relative percentages of the peak areas according to ER = E1/E2, where [E1] and [E2] are the peak areas for each enantiomer [43].

### 4.2. Cell Culture

The hnMSCs isolated from human umbilical cord stroma were fully characterized as described elsewhere [44] and expanded in Eagle’s minimum essential medium—alpha modification (α-MEM) supplemented with 10% (*v*/*v*) of fetal bovine serum (FBS; Gibco, Paisley, UK) [23]. For generating HLCs, a three-step differentiation protocol was applied to hnMSCs, as detailed previously [19]. Briefly, hnMSCs were seeded at a density of 1.5 × 10^5^ cells/cm^2^ in a rat-tail collagen (0.2 mg/mL) coated surface plates. At day 17 (D17) of differentiation, cells were trypsinized and reinoculated in Iscove’s modified Dulbecco’s medium (IMDM) with 8 ng/mL oncostatin M (OSM; Peprotech, Rocky Hill, NJ, USA), 1 µM dexamethasone, 1% dimethyl sulfoxide (DMSO) and 1% insulin-transferrin-selenium solution (Gibco, Grand Island, NY, USA), 20 µM of 5-azacytidine, and 5% (*v*/*v*) FBS into (i) ultra-low attachment (ULA; VWR International, Radnor, PA, USA) plates (5 × 10^5^ cells/mL) to obtain a 3D spheroid culture or (ii) 2D culture plates pre-coated with collagen (2 × 10^4^ cells/cm^2^). Medium was changed 24 h after inoculation to remove 5-azacytidine and FBS. Cells were maintained until D34 in this medium, which was replaced every 3 days.

### 4.3. Cell Viability Assessment

The cytotoxicity of the racemic mixtures (0.25, 0.5, 1.0, 2.0, 3.0, 4.0, 6.0, 8.0, 10.0, and 20.0 mM) and enantiomers (0.5 and 1.0 mM) of pentedrone and methylone were evaluated in both 2D monolayer and 3D spheroid cultures of HLCs, maintained as described above with slight changes. Briefly, at D17 of differentiation, cells were seeded in 96-well plates in 200 μL of culture medium at a density of 6.4 × 10^3^ cells/well and 1.6 × 10^4^ cells/well in 2D and 3D cultures, respectively. At D34, HLCs cultured in 2D and 3D were exposed to the abovementioned racemic mixtures and enantiomers diluted in IMDM. Cell viability was measured upon 24 h of incubation with the compounds using the CellTiter-Glo^®^ 3D Cell Viability Assay (Promega, Madison, WI, USA), according to manufacturer instructions. Five percent DMSO was used as negative control and IMDM as positive control. IC_50_ was calculated for the racemic mixtures using a nonlinear regression fit for the Log_10_ transformation of the concentration values by using GraphPad Prism version 7.0 (GraphPad Software, La Jolla, CA, USA). Three independent experiments were performed.

### 4.4. In Vitro Metabolism Assays

The metabolism of methylone and pentedrone was initially evaluated by incubation with ultra-pool human liver microsomes (HLM) at a final assay concentration of 1 mg protein/mL. All experiments were performed in duplicate. The incubation mixture with the HLM consisted of 1.3 mM NADP^+^, 3.3 mM glucose-6-phosphate, 3.3 mM MgCl_2_, 0.4 IU/mL glucose-6-phosphate dehydrogenase, and 0.05 mM sodium citrate in 0.1 M phosphate buffer at pH 7.4. The incubation mixture with each drug was pre-incubated 10 min at 37 °C before the experiment was started with the addition of the HLM. All microsomes assays were incubated with 50 µM of each enantiomer in a final volume of 1 mL at 37 °C. At the time points 0, 5, 10, 20, 30, 40, 50, and 60 min, 100 µL aliquots of the incubations were quenched with 100 µL of ice-cold acetonitrile and methcathinone (internal standard) 100 µg/mL. The quenched solutions were centrifuged for 10 min at 4 °C and the supernatants were extracted with ethyl acetate after previous pH correction (pH = 11 for pentedrone and pH = 13 for methylone).

Regarding the study of methylone and pentedrone metabolite biosynthesis, in 2D-HLCs and 3D-HLCs, at the D17 of the HLCs differentiation, the cells were seeded in 6-well plates in 1 mL of culture medium at a density of 2.0 × 10^5^ cells/well and 5.0 × 10^5^ cells/well in 2D and 3D cultures, respectively. At D34, 2D-HLCs and 3D-HLCs were exposed to 1 mM of each enantiomer diluted in IMDM. Upon 6 h of incubation, the IMDM was removed and the pellets were quenched with 2 mL of ice-cold acetonitrile and scraped. The quenched solutions were centrifuged for 10 min at 4 °C. After the addition of methcathinone (internal standard) 100 µg/mL, the supernatants were derivatized with TFAA and analyzed by GC–MS [2,41].

### 4.5. Synthesis of Dihydromethylone and Dihydropentedrone

The reagents and solvents, sodium borohydride, sodium hydroxide, ethanol, and ethyl acetate, were purchased from Sigma-Aldrich Co (St. Louis, MO, USA), and were used without further purification. Silica gel 60 GF_254_ (Merck, Darmstadt, Germany) precoated plates were used for TLC and ^1^H NMR was recorded on a Brucker DRX-300 spectrometer in CDCl_3_ at room temperature. Chemical shifts are expressed in δ (ppm) values relative to TMS. IR spectra were obtained on an FTIR spectrometer Nicolet iS10 from Thermo Scientific (Waltham, MA, USA) with Smart OMNI-Transmission accessory (Software 188 OMNIC 8.3) in KBr.


*General procedure for the synthesis of metabolites of cathinones by reduction of keto group.*


To a solution of pentedrone (200 mg, 1.0 mmol) or methylone (207 mg, 1.0 mmol) in ethanol (10 mL), stirred and cooled in an ice bath, sodium borohydride (118 mg, 3.0 mmol) was added, in one portion. The ice bath was removed after 15 min and the reaction was stirred at room temperature for 2 h. Then, the solvent was evaporated. The residue was diluted with water (10 mL), 1 M sodium hydroxide was added to adjust pH to 12–14, and the mixture was extracted with ethyl acetate (4 × 20 mL). The combined organic layers were dried with anhydrous sodium sulphate, filtered, and the solvent evaporated to afford the reduced products, as white solids.

Dihydropentedrone was obtained as white crystals (Yield: 63.7%). IR ν_max_ (cm^−1^) (KBr): 3250–2753, 2975, 1475, 1050, 810, 785, 733; ^1^H NMR (300.13 MHz, CDCl_3_) δ: 7.34 (d, J = 6.6 Hz, 2H, H-2′ and H6′), 7.32 (t, J = 6.6 Hz, 1H, H-4′), 7.26 (t, J = 6.6 Hz, 2H, H-3′ and H-5′), 4.85 (d, J = 3.9 Hz, 1H, H-1), 2.63 (m, 1H, H-2), 2.52 (s, 3H, CH_3_), 1.30 (m, 2H, H-3), 1.20 and 1.12 (2×m, 2H, H-4), 0.89 ppm (t, J = 7.0 Hz, 3H, H-5); ^13^C NMR (75.47 MHz, CDCl_3_) δ: 141.5 (C-1′), 128.2 (C-3′ and C-5′), 127.1 (C-2′ and C-6′), 126.2 (C-4′), 71.8 (C-1), 65.2 (C-2), 34.8 (CH_3_), 30.5 (C3), 19.8 (C-4), 14.2 (C-5); Retention time (RT) in GC–MS: 7.13 min. Chemical equation (with TFAA): C_16_H_17_F_6_NO_3_ *m*/*z* 385. Fragments observed with GC–MS (listed in order of decreasing abundance) *m*/*z* 140 (C_4_H_5_F_3_NO); 182 (C_7_H_11_F_3_NO); 203 (C_9_H_6_F_3_O_2_); 385 (C_16_H_17_F_6_NO_3_); 272 (C_14_H_17_F_3_NO).

Dihydromethylone was obtained as white oil (Yield: 96.5%). IR ν_max_ (cm^−1^) (KBr): 3248-2750, 2893, 1485, 1440, 1245, 1085, 932, 810, 785, 733; ^1^H NMR (300.13 MHz, CDCl_3_) δ: 6.86 (s, 1H, H-2′), 6.78 (d, J = 8.7 Hz, 1H, H-5′), 6.75 (d, J = 8.7 Hz, 1H, H-6′), 5.94 (s, 2H, CH_2_), 4.79 (brs, 1H, H-1), 2.81 (m, 1H, H-2), 2.51 (s, 3H, CH_3_), 0.89 ppm (d, J = 3.8 Hz, 3H, H-3); ^13^C NMR (75.47 MHz, CDCl_3_) δ: 147.7 (C-3′), 146.7 (C-4′), 135.4 (C-1′), 119.2 (C-6′), 108.1 (C-2′), 106.9 (C-5′), 101.0 (CH_2_), 72.7 (C-1), 60.8 (C-2), 29.8 (CH_3_), 14.3 (C3); RT in GC–MS: 9.54 min. Chemical equation (with TFAA): C_15_H_13_F_6_NO_5_ *m*/*z* 401. Fragments observed with GC–MS (listed in order of decreasing abundance) *m*/*z* 154 (C_5_H_7_F_3_NO); 247 (C_10_H_6_F_3_O_4_); 121 (C_7_H_5_O_2_); 401 (C_15_H_13_F_6_NO_5_).

As supplementary data, GC–MS spectra of dihydropentedrone (Appendix A) and dihydromethylone (Appendix A) in (A) full scan and B) SIM mode are available, as well as ^1^H NMR (300.13 MHz, CDCl_3_) and ^13^C NMR (75.48 MHz, DMSO-d_6_) for dihydromethylone (Appendix A) and dihydropentedrone (Appendix A). 

### 4.6. Identification of Pentedrone and Methylone Metabolites by GC–MS Analysis

To study pentedrone and methylone metabolic profiles, we set up a methodology for the identification of pentedrone and methylone phase I metabolites based on a GC–MS methodology developed by Araujo et al. and Uralets et al. [2,41] with some adaptations. Briefly, samples from the metabolic assays (200 μL) were evaporated to dryness under nitrogen flow. Then, 50 µL of ethyl acetate and 50 µL of TFAA (derivatizing agent) were added to the dried residue. Incubation was performed at 65 °C for 15 min. After cooling to room temperature, the extract was dried under nitrogen flow. The residue obtained was dissolved in 100 μL of ethyl acetate, and 2 μL was injected into the GC–MS apparatus.

GC–MS analysis was performed with a 436-GC model (Bruker Daltonics, Billerica, MA, USA) coupled to an EVOQ triple quadrupole mass spectrometer (Bruker Daltonics) and a Bruker MS workstation software version 8.2. The chromatographic separation was accomplished using a fused silica capillary column (Rxi-5Sil MS; 30 m × 0.25 mm × 0.25 μm; Restek Corporation, U.S., Bellefonte, PA, USA). The carrier gas was helium C-60 (Gasin, Portugal), at a constant flow of 1 mL/min. Injections were performed in split mode, with a ratio of 1/20. The injector port was heated to 250 °C. The initial column temperature was 80 °C, followed by a temperature increase of 15 °C/min to 150 °C and 10 °C/min to 300 °C. The MS detector was operated in EI mode (70 eV). The temperature of the transfer line, manifold and EI were 280, 40, and 270 °C, respectively. Data acquisition was performed in full scan mode with a mass range between 50 and 800 *m*/*z* at a scan rate of 8 scans/s. The emission current was 40 µA. Ionization was maintained off during the first 1.5 min to avoid solvent overloading. Total separation run time was 20 min.

The analysis was initially performed in full scan mode. However, the acquisition of all mass spectra of the complete chromatogram (full scan mode) did not guarantee the identification of all peaks in each chromatogram due to the presence of some chromatographic peaks (reminiscent sugars, for example) that interfere with the analysis. For that, specific metabolites of methylone and pentedrone, already identified in previous studies [2,15], were selected to postulate their mass fragmentation according to their chemical structures (Figure 11) for posterior analysis using the specific ions (*m*/*z* in SIM mode) to identify and confirm the metabolite peaks. In SIM mode the selected *m*/*z* ions were: IS—*m*/*z* 77, 105, 110, and 154; methylone—*m*/*z* 121, 149, 154, and 303; nor-methylone—*m*/*z* 121, 140, 149, 168, 289, and 260; dihydromethylone—*m*/*z* 121, 154, 247, 280, and 401; DHMC—*m*/*z* 154, 182, 209, 237, and 391; *N*-hydroxymethylone—*m*/*z* 121, 149, 170, 198, and 319; nor-pentedrone—*m*/*z* 77, 105, 126, 168, and 273; dihydropentedrone—*m*/*z* 140, 182, 203, 272, and 385; and pentedrone—*m*/*z* 77, 105, 140, and 182.

## 5. Conclusions

Overall, the present work provided evidence of the enantioselectivity for pentedrone and methylone in the metabolic processes in the liver, using more physiological in vitro models. In fact, the enantioselectivity of pentedrone and methylone differ according to the metabolic pathway. A different, but nonetheless complementary, metabolic profile between HLCs cultured in 2D and 3D was achieved, sustaining that the model system must be chosen according to the objective of the study. These results may help to understand liver damage reported in humans. Moreover, this study revealed for the first time enantioselectivity in cytotoxicity for pentedrone. Nevertheless, further studies are needed to disclose the role of enantioselectivity in the underlying mechanisms toxicity of these psychoactive substances.

## Figures and Tables

**Figure 1 pharmaceuticals-15-00368-f001:**
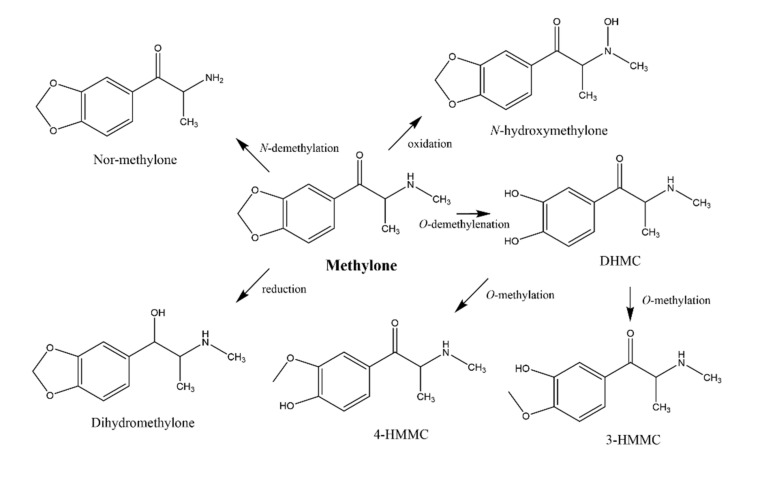
In vitro and in vivo metabolism of methylone. DHMC: dihydroxymethcathinone; 4-HMMC: 4-hydroxy-3-methoxy-*N*-methylcathinone; 3-HMMC: 3-hydroxy-4-methoxy-*N*-methylcathinone (adapted from [11,13,14,15,16]).

**Figure 2 pharmaceuticals-15-00368-f002:**
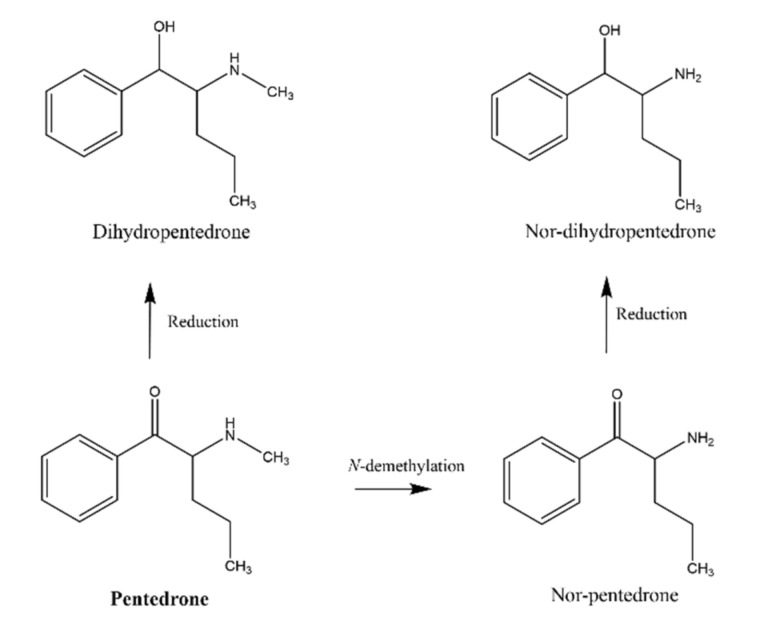
Metabolism of pentedrone in human urine (adapted from [2]).

**Figure 3 pharmaceuticals-15-00368-f003:**
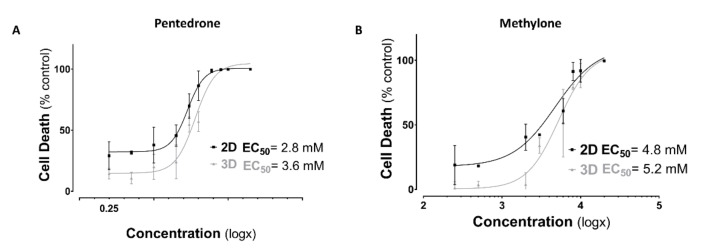
Nonlinear regression models of dose–response curves in 2D and 3D HLCs cells 24 h after exposure to (**A**) pentedrone and (**B**) methylone (0–20.0 mM) racemates. Data are presented as mean ± SD. Concentration–response curves were fitted using least squares as the fitting method and the comparisons between curves were made using the extra sum-of-squares F test. Results were obtained from three independent experiments, performed in quadruplicate.

**Figure 4 pharmaceuticals-15-00368-f004:**
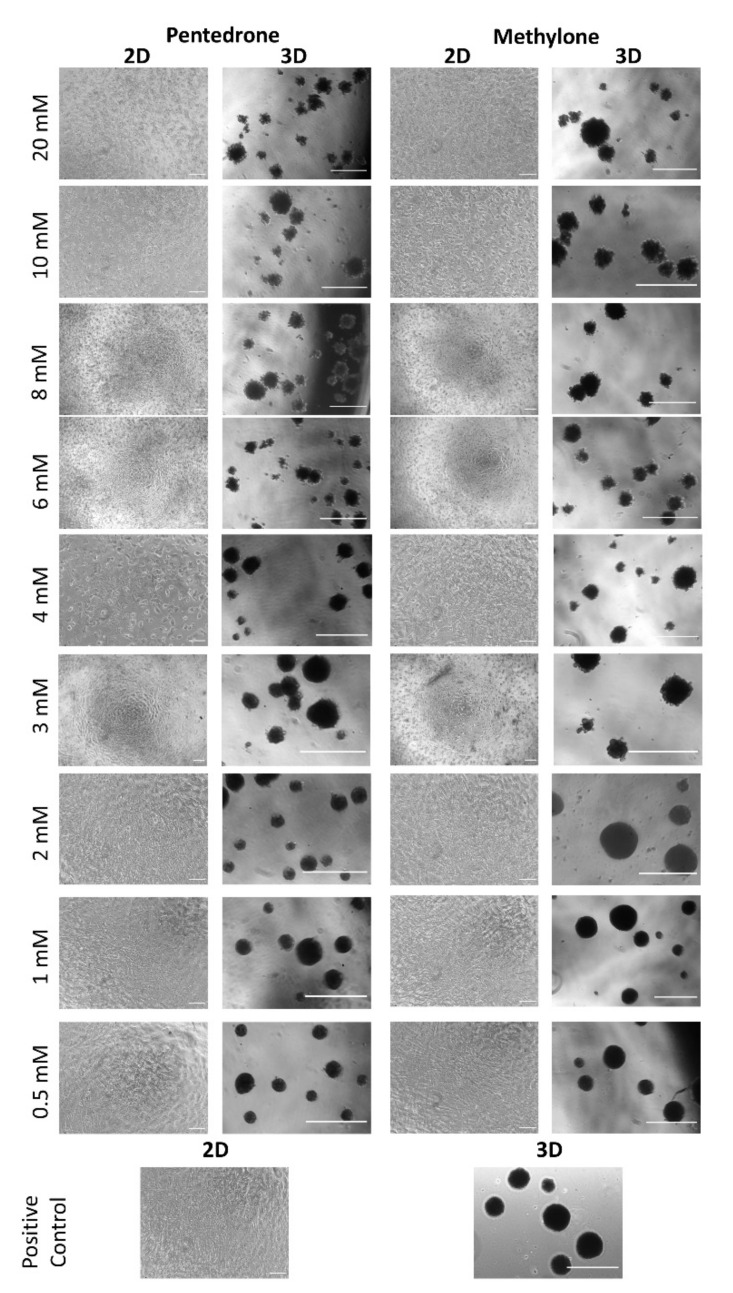
Representative images of HLCs’ morphology in 2D (scale bar = 100 µm) and 3D (scale bar = 200 µm) cultures exposed and nonexposed (positive controls) to pentedrone and methylone racemates.

**Figure 5 pharmaceuticals-15-00368-f005:**
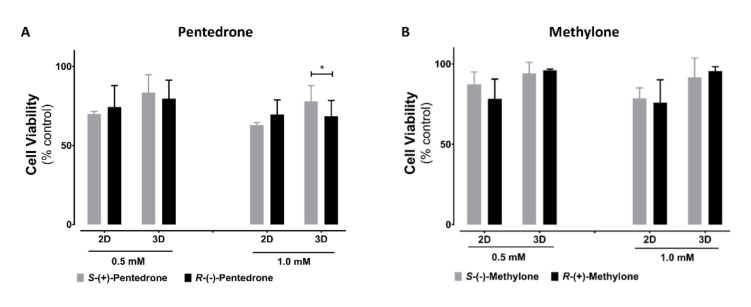
Cytotoxicity in HLCs exposed to 0.5 and 1.0 mM of (**A**) pentedrone and (**B**) methylone enantiomers evaluated by CellTiter-Glo^®^, after 24 h. Data are presented as average ± SD (*n* = 3). * *p* < 0.05.

**Figure 6 pharmaceuticals-15-00368-f006:**
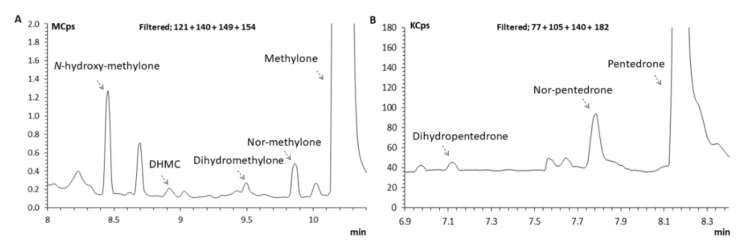
Zoomed full scan chromatogram of an extracted sample of HLM incubated with *S*-(−)-methylone and *S*-(+)-pentedrone for 60 min reconstructed with the following ions: (**A**) *m*/*z* = 121, *m*/*z* = 149, *m*/*z* = 140, and *m*/*z* = 154, *N*-hydroxymethylone (8.45 min), DHMC (8.95 min), dihydromethylone (9.50 min), nor-methylone (9.86 min), and methylone (10.19 min); (**B**) *m*/*z* = 77, *m*/*z* = 105, *m*/*z* = 140, and *m*/*z* = 182, dihydropentedrone (7.14 min), nor-pentedrone (7.76 min), and pentedrone (8.12 min).

**Figure 7 pharmaceuticals-15-00368-f007:**
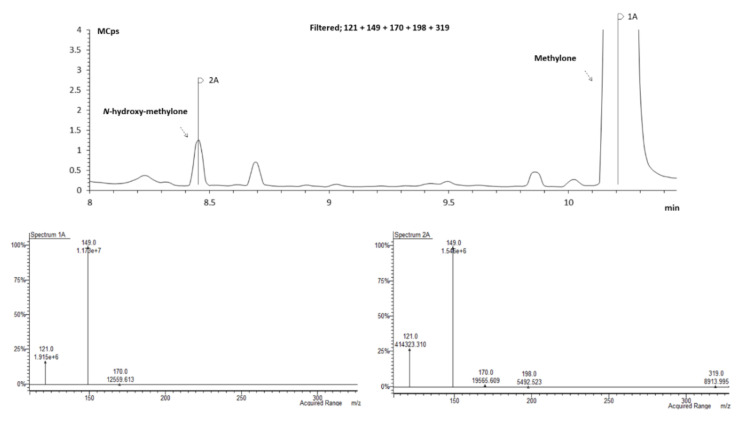
Zoomed GC–MS SIM mode chromatogram of an extracted sample of HLM incubated with *S*-(−)-methylone for 30 min.

**Figure 8 pharmaceuticals-15-00368-f008:**
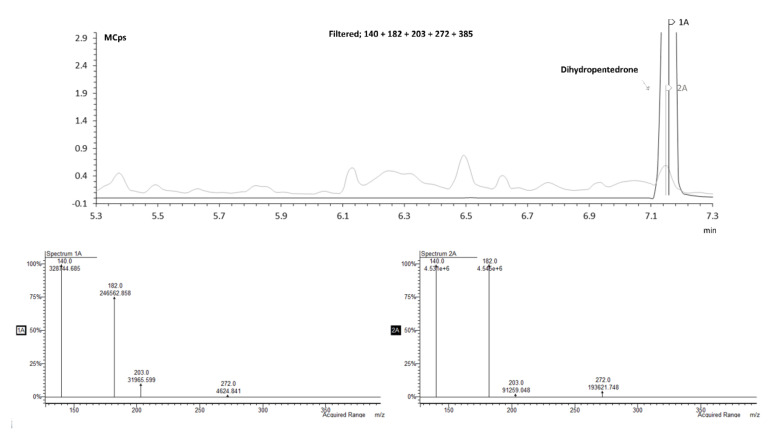
Overlapped zoomed GC–MS SIM mode chromatogram of an extracted sample of HLM incubated with *S*-(+)-pentedrone for 20 min (spectrum 1A) and dihydropentedrone (spectrum 2A).

**Figure 9 pharmaceuticals-15-00368-f009:**
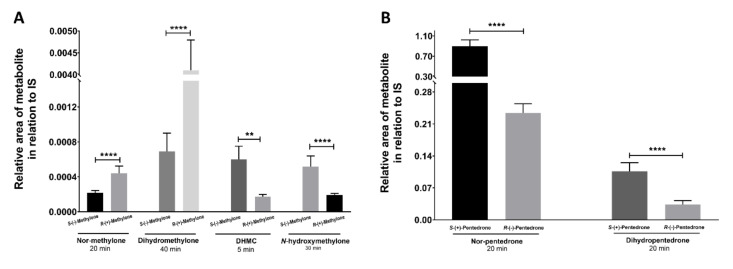
(**A**) Methylone metabolites (nor-methylone, DHMC, *N*-hydroxymethylone, and dihydromethylone) produced in the peak time points after incubation with microsomes (1 mg/mL). (**B**) Pentedrone metabolites (nor-pentedrone and dihydropentedrone) produced in specific time points after incubation with microsomes (1 mg/mL). Data are presented as average ± SD (*n* = 2). ** *p* < 0.01, **** *p* < 0.0001.

**Figure 10 pharmaceuticals-15-00368-f010:**
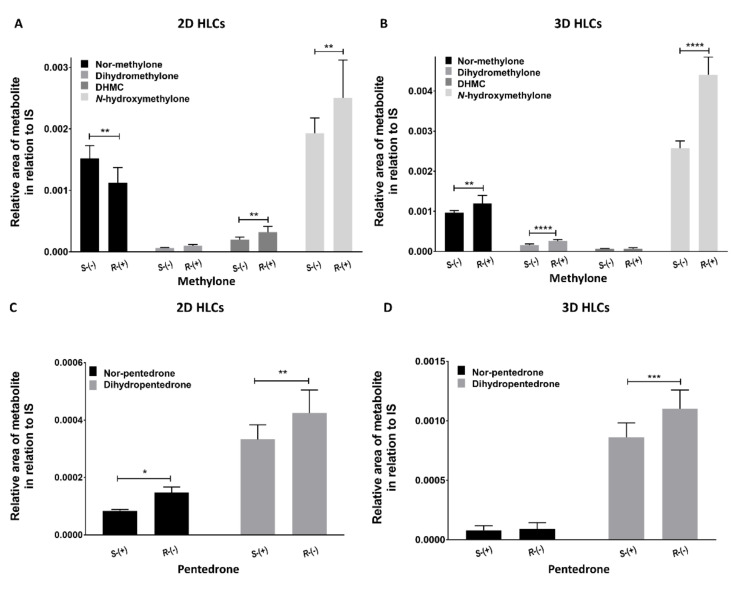
Metabolic profile of (**A**) methylone in 2D-HLCs, (**B**) methylone in 3D-HLCs, (**C**) pentedrone in 2D-HLCs, and (**D**) pentedrone in 3D-HLCs after 6 h of exposition. Data are presented as average ± SD (*n* = 2). * *p* < 0.05, ** *p* < 0.01, *** *p* < 0.001, **** *p* < 0.0001.

**Figure 11 pharmaceuticals-15-00368-f011:**
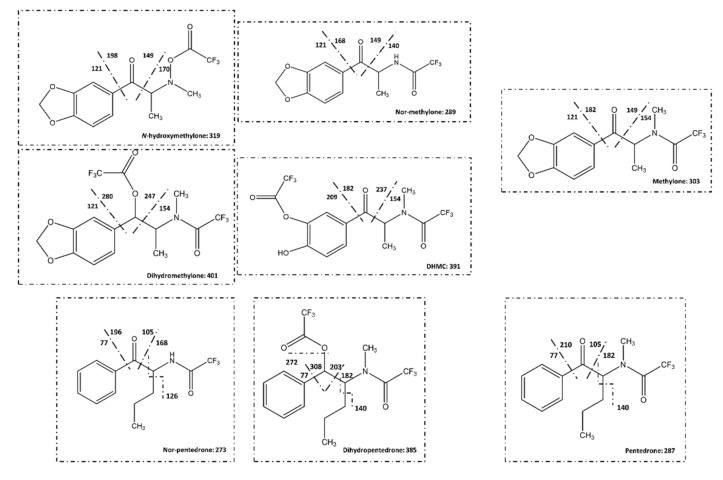
Mass fragmentation of methylone, pentedrone, and their metabolites. DHMC: dihydroxymethcathinone.

## Data Availability

Data is contained within the article and supplementary material.

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
