# Peer review of "Enantioselectivity of Pentedrone and Methylone on Metabolic Profiling in 2D and 3D Human Hepatocyte-like Cells"

_pharmaceuticals, 2022, doi:10.3390/ph15030368_

Round 1

Reviewer 1 Report

This manuscript describes an interesting evaluation of the enantioselective metabolic profile of pentedrone and methylone, carried out on an hepatocyte model cultured under 3D and 2D conditions. The experiments were carried out through well established methodologies and reported with  good clarity and sufficient details. The results were presented accurately, with a proper statistical analysis. Overall, I repute this study interesting for the readers of this journal; anyway, I suggest the following minor modifications to improve the manuscript:

1) the quality of figures illustrating chemical structures is too low. Molecules are depicted using different styles (figs 1, 2 and 11). Please unify the style of chemical drawing and improve the resolution of figures.

2) the use of 3D and 2D HLC models should be better explained by authors, to help the readers  not experienced in this specific field to have a more deepen understanding.

3) In the synthesis description (paragraph 4.5) it is not mentioned if compounds have been purified. Please specify it, including conditions of purification. In addition, the description of chemical characterization of compounds should include 13C spectra and high resolution mass spectra (or elemental analysis). I think that the current description (with only 1H spectrum and GC/MS) it is not a sufficient evaluation.

4) the title of paragraph 4.6 is wrong. Please correct it.

Author Response

We thank very much the reviewers’ suggestions and corrections to our paper. On behalf of my co-authors, I am re-submitting the review manuscript entitled “Enantioselectivity of pentedrone and methylone on metabolic profiling in 2D and 3D human hepatocyte-like cells” for publication in Pharmaceuticals Journal, after it has been carefully and thoroughly reviewed and corrected according to the reviewers’ suggestions and observations. 

All changes introduced in the text are identified in the track changes tool.

REVIEWER 1

This manuscript describes an interesting evaluation of the enantioselective metabolic profile of pentedrone and methylone, carried out on an hepatocyte model cultured under 3D and 2D conditions. The experiments were carried out through well established methodologies and reported with  good clarity and sufficient details. The results were presented accurately, with a proper statistical analysis.

We acknowledge the comments made by the reviewer.

Overall, I repute this study interesting for the readers of this journal; anyway, I suggest the following minor modifications to improve the manuscript: 1) the quality of figures illustrating chemical structures is too low. Molecules are depicted using different styles (figs 1, 2 and 11). Please unify the style of chemical drawing and improve the resolution of figures.

We agreed with the reviewer’s suggestion. The change and improvement of the figures 1, 2 and 11 was done and replaced in the manuscript.

2) the use of 3D and 2D HLC models should be better explained by authors, to help the readers  not experienced in this specific field to have a more deepen understanding.

We agreed with the reviewer’s suggestion. The following text was added to the introduction section: …“Traditionally, such cultures are carried out under adherent conditions wherein the cells are attached to a glass or plastic dish, designated as 2D culture models. The advantages of 2D cultures are several and are mostly associated with simple and low-cost maintenance of the cell culture and with the performance of functional tests. However, 2D models also have numerous disadvantages. Adherent cell cultures do not mimic the natural structures of tissues, namely cell-cell and cell-extracellular matrix (ECM) interactions responsible for cell differentiation, proliferation, vitality, expression of genes and proteins, responsiveness to stimuli, drug metabolism and other cellular functions. Therefore, under such conditions cells morphology and phenotype is altered, affecting their function, structural organization, secretion and cell signalling. Owing to the many disadvantages of 2D systems, major efforts have been made to provide more physiologically relevant models for non-clinical hepatotoxicity assessment. 3D models are more complex culture systems where multicellular microphysiological devices are explored within a vision to replicate the characteristics and response of human tissues in vivo. 3D culture systems enable cell polarization, the formation of nutrient and oxygen gradients, and higher cell-cell and cell-ECM contacts, which are key conditions for the maintenance of hepatic-specific functions [18-20]. We have previously developed a hepatic differentiation protocol for obtaining HLCs from human mesenchymal stem cells (hnMSCs) [21-23]. The hnMSCs are a promising stem cell type, regarding accessibility, expandability, ethical and safety concerns, when compared to human-induced pluripotent stem cells (hiPSCs) or human embryonic stem cells (hESCs).”…

3) In the synthesis description (paragraph 4.5) it is not mentioned if compounds have been purified. Please specify it, including conditions of purification. In addition, the description of chemical characterization of compounds should include 13C spectra and high resolution mass spectra (or elemental analysis). I think that the current description (with only 1H spectrum and GC/MS) it is not a sufficient evaluation.

After the synthesis and multiple liquid-liquid extraction of both reduced cathinones, a white solid was obtained (we added this information in the manuscript). In this study, the synthesis of these two compounds was performing aiming to afford the standards of the reduced metabolites of pentedrone and methylone to validate the identification of GC-MS method used for the determination of pentedrone and methylone metabolites (that was based on spectra comparison and time-course formation of metabolites). We thank and agree with reviewer’s suggestion and include the 13C NMR data of both compounds. Considering the purpose of synthetize these compounds, we consider that the characterization of compounds by IR, 1H  NMR, 13C NMR and GC/MS is sufficient. The 1H NMR, 13C NMR and GC/MS spectra of the synthesized compounds were included in the Supplementary Materials.

4) the title of paragraph 4.6 is wrong. Please correct it.

We thank the reviewer for the comment, the title of 4.6 was changed to “Pentedrone and methylone metabolites identification by GC-MS analysis”.

Reviewer 2 Report

The manuscript "Enantioselectivity of pentedrone and methylone on metabolic 2 profiling in 2D and 3D human hepatocyte-like cells" assessed both hepatic cytotoxic and metabolic profile of pentedrone and methylone enantiomers using physiologically relevant in vitro models. The subject reveals a distinct difference of pentedrone and methylone in both hepatic cytotoxic and metabolic profiles for their enantiomers, which provided a certain reference value for related toxicology and pharmacology research. The subject matter of this article is in line with the scope of Pharmaceuticals. I recommend acceptance of the manuscript after some revisions for the following reasons:
1. In figure 1 and figure 2, the stereo structure of compounds needs to be described by wedged bond.
2. The NMR spectra of synthesized compounds need to be attached in Supplementary Materials.
3. Chromatographic conditions for isolation of chiral compounds using chiral columns should be described in "Materials and Methods".

Author Response

We thank very much the reviewers’ suggestions and corrections to our paper. On behalf of my co-authors, I am re-submitting the review manuscript entitled “Enantioselectivity of pentedrone and methylone on metabolic profiling in 2D and 3D human hepatocyte-like cells” for publication in Pharmaceuticals Journal, after it has been carefully and thoroughly reviewed and corrected according to the reviewers’ suggestions and observations. 

All changes introduced in the text are identified in the track changes tool.

REVIEWER 2

The manuscript "Enantioselectivity of pentedrone and methylone on metabolic 2 profiling in 2D and 3D human hepatocyte-like cells" assessed both hepatic cytotoxic and metabolic profile of pentedrone and methylone enantiomers using physiologically relevant in vitro models. The subject reveals a distinct difference of pentedrone and methylone in both hepatic cytotoxic and metabolic profiles for their enantiomers, which provided a certain reference value for related toxicology and pharmacology research. The subject matter of this article is in line with the scope of Pharmaceuticals. I recommend acceptance of the manuscript after some revisions for the following reasons:

We acknowledge the comments made by the reviewer.

  1. In figure 1 and figure 2, the stereo structure of compounds needs to be described by wedged bond.

The figures 1 and 2 show the described metabolism of methylone and pentedrone as racemates (with no configuration assignment). The metabolism of each enantiomer of these drugs is not described in the literature. One novelty of this manuscript is the first report of enantioselectivity of these compounds on the metabolism.

  1. The NMR spectra of synthesized compounds need to be attached in Supplementary Materials.

We agreed with the reviewer’s suggestion, NMR spectra of the synthesized compounds were included in the Supplementary Materials.

  1. Chromatographic conditions for isolation of chiral compounds using chiral columns should be described in "Materials and Methods".

The following text was added to a subsection 4.1.1: “The enantiomers of pentedrone and methylone were isolated by LC on a Chiralpak AS® (250 × 10 mm i.d., 5 μm particles size) column, as previously described [26]. Briefly, multiple injections (500 μL loop) using Hex:2-PrOH as mobile phase were performed. Analyses were performed at 25°C, in isocratic mode under UV detection (254 nm). The detection parameters were adjusted to a peak start and end threshold of 200 mAU for pentedrone enantiomers and 100 mAU for methylone. The obtained fractions of each enantiomer were precipitated with HCl on diethyl ether (2M). Working solutions for each enantiomer of pentedrone and methylone were prepared at a concentration of 0.1 mg/mL and injected, in triplicate, on the analytical Chiralpak AS-H® (15 cm x 4.6 mm i.d., 5 μm particles size) column under the optimized chromatographic conditions, as previously described [28] to measure their enantiomeric purity (e.r.). The enantiomeric ratio (ER) was determined by the relative percentages of the peak areas according to ER = E1/E2, where [E1] and [E2] are the peak area of each enantiomer [43].”.